# Unveiling the Mechanism of Action of 7α-acetoxy-6β-hydroxyroyleanone on an MRSA/VISA Strain: Membrane and Cell Wall Interactions

**DOI:** 10.3390/biom10070983

**Published:** 2020-06-30

**Authors:** Filipe Pereira, Teresa Figueiredo, Rodrigo F. M. de Almeida, Catarina A. C. Antunes, Catarina Garcia, Catarina P. Reis, Lia Ascensão, Rita G. Sobral, Patricia Rijo

**Affiliations:** 1Research Center for Biosciences and Health Technologies (CBIOS), Universidade Lusófona de Humanidades e Tecnologias, Campo Grande, 376, 1749-024 Lisboa, Portugal; pereira.filipepereira@gmail.com (F.P.); catarina.g.garcia@gmail.com (C.G.); catarinareis@ff.ulisboa.pt (C.P.R.); 2Department of Biomedical Sciences, Faculty of Pharmacy, University of Alcalá, Campus Universitario, 28871 Alcalá de Henares, Spain; 3Departamento de Ciências da Vida, UCIBIO@REQUIMTE, Faculdade de Ciências e Tecnologia, Universidade Nova de Lisboa, 2825-149 Caparica, Portugal; figteresa@gmail.com; 4Centro de Química Estrutural, Faculdade de Ciências, Universidade de Lisboa, 1749-016 Lisboa, Portugal; rfalmeida@fc.ul.pt (R.F.M.d.A.); catarina.alves.antunes@gmail.com (C.A.C.A.); 5Instituto de Investigação do Medicamento (iMed.ULisboa), Faculdade de Farmácia, Universidade de Lisboa, 1649-003 Lisboa, Portugal; catarinareis@ff.ulisboa.pt; 6Institute of Biophysics and Biomedical Bioengeneering (IBEB), Faculdade de Ciências, Universidade de Lisboa, 1749-016 Lisboa, Portugal; 7Centro de Estudos do Ambiente e do Mar (CESAM), Faculdade de Ciências, Universidade de Lisboa, 1749-016 Lisboa, Portugal; lmpsousa@fc.ul.pt

**Keywords:** 7α-Acetoxy-6β-hydroxyroyleanone, MRSA, antibacterial activity, membrane interaction, cell wall

## Abstract

The number of cases of failure in the treatment of infections associated with resistant bacteria is on the rise, due to the decreasing efficacy of current antibiotics. Notably, 7α-Acetoxy-6β-hydroxyroyleanone (AHR), a diterpene isolated from different *Plectranthus* species, showed antibacterial activity, namely against Methicillin-resistant *Staphylococcus aureus* (MRSA) strains. The high antibacterial activity and low cytotoxicity render this natural compound an interesting alternative against resistant bacteria. The aim of this study is to understand the mechanism of action of AHR on MRSA, using the MRSA/Vancomycin-intermediate *S. aureus* (VISA) strain CIP 106760, and to study the AHR effect on lipid bilayers and on the cell wall. Although AHR interacted with lipid bilayers, it did not have a significant effect on membrane passive permeability. Alternatively, bacteria treated with this royleanone displayed cell wall disruption, without revealing cell lysis. In conclusion, the results gathered so far point to a yet undescribed mode of action that needs further investigation.

## 1. Introduction

Due to the continuously increasing resistance of bacteria against front-line antibiotics, numerous attempts to develop new antibacterial agents effective against disseminating infections have been made [1].

Multidrug resistant strains such as Methicillin-resistant *Staphylococcus aureus* (MRSA) have been spreading at alarming rates. Adverse outcomes, such as associated increased morbidity and mortality, are concern priorities for healthcare policies. Surveillance and control strategies are necessary interventions towards a decreased prevalence of such pathogens [1]. However, discovering and/or developing new antibacterial agents, with alternative mechanisms of action, is an essential field for further study.

The treatment of human infections resorting to plants is long established in traditional medicine. The demonstrated therapeutic effects of some plants have attracted researchers to discover which compounds are responsible for such bioactivity [2].

Secondary metabolites, such as diterpenes, have been the focus of a series of scientific studies due to their acknowledged bioactivity [3]. Diterpenes are important scaffolds for antibacterial properties, where the length and flexibility of the alkyl chain with carbonyl groups are crucial factors to increase the antimicrobial activity of molecules. Many royleanone diterpenes, including 7α-acetoxy-6β-hydroxyroyleanone (AHR), are documented for their ascertained antibacterial activity [2,4]. AHR is frequently found in *Plectranthus grandidentatus* Gürke (Lamiaceae), and its optimized isolation has been documented [4]. Along with other similar diterpenes, this compound has revealed activity against Gram-positive bacteria, and more importantly, against MRSA strains [2,5,6].

However, to the best of our knowledge, its mechanism of action is not entirely known. It is described that these diterpenes have the ability to transverse or damage the bacterial cytoplasmatic membrane due to their amphipathic character, and that the antibacterial activity may be modulated through an increase in lipophilicity and/or in hydrogen-bond donor/acceptor groups of the hydrophilic moiety [2]. In the hope of unveiling the mechanism behind bacterial death, the effect of AHR on the MRSA bacterial cell membrane and cell wall is herein studied.

## 2. Materials and Methods

### 2.1. Reagents

First, 1,2-dioleoyl-*sn*-glycero-3-phosphocholine (DOPC), 1-palmitoyl-2-oleoyl-*sn*-glycero-3-phosphocholine (POPC) and *N*-palmitoyl-sphingomyelin (PSM) were purchased from Avanti Polar Lipids (Alabaster, AL). Cholesterol, 5,6-carboxyfluorescein (CF), 8-hydroxypyrene-1,3,6-trisulfonic acid (HPTS) trisodium salt (pyranine), Sephadex G-25 and Triton X-100 were acquired from Sigma (Sintra, Portugal). Solvents for lipid and Royleanone derivative (AHR) stock solutions were of spectroscopic grade. Methicillin, ampicillin and vancomycin were obtained from Sigma. 

### 2.2. Extraction, Isolation and Purification of AHR

Three extraction processes were performed, to identify which extraction method is the one with higher yield of AHR from *Plectranthus grandidentatus* Gurke [4,7]. The plant material was used as previously described [4,7]. Supercritical fluid (SCF) extraction was carried out in an experimental apparatus equipped with a sample cell with 100 dm^3^. Air-dried leaves (100.45 g) were powdered extracted with CO_2_ for 4 h at 40 °C and 230 bar. After extraction, the extract was recovered with acetone. Acetone maceration extraction was carried out using 100.88g of dried plant and 1L acetone at room temperature and strong agitation for 1h. The resulting extract was then filtrated, and the solvent was evaporated in a rotary evaporator. Acetone ultrasonic-assisted extraction was carried out using 100.13 g of dried powdered leaves with 1L acetone. The suspension was maintained in an ultrasonic bath for 1h at room temperature. After extraction, the suspension was filtrated and the solvent was evaporated. The isolation and consequently structural identification of AHR was conducted through the different chromatographic methods previously described. Thus, AHR was structurally identified and confirmed by spectroscopic means, according to the literature [5,7,8,9,10,11]. The content of AHR present in the extract was evaluated by high-pressure liquid chromatography-diode-array detection (HPLC-DAD), according to the method developed and published previously [12].

### 2.3. Liposome Preparation

To obtain a final lipid concentration of 5 mM in the liposomal suspensions, the appropriate volume of a stock solution of lipid in chloroform was mixed and the solvent was removed by evaporation under a mild flow of nitrogen. Then, samples were placed in a vacuum overnight to ensure the complete elimination of organic solvent. The lipid film was hydrated with 10 mM Tris-HCl, 20mM NaCl 0.1 mM EDTA buffer, to a final lipid concentration of 5 mM. After that, seven freeze/thaw cycles (liquid nitrogen/water bath) were performed. Subsequently, 100 nm diameter large unilamellar vesicles (LUVs) were prepared by the extrusion method as described previously [7]. 

The stock solution of AHR (10 mg/mL) was prepared in dimethyl sulfoxide (DMSO) and stored at −18 °C. During each experiment, a fresh aliquot solution was used and diluted in the corresponding buffer. The concentration of DMSO on experiment solution was under 1% (*v/v*).

### 2.4. Absorption Measurements

The absorption spectra were obtained with a Shimadzu UV 560 double beam spectrophotometer, in the absence and presence of LUV suspension, in Tris-HCl, pH 7.4 buffer. DOPC LUVs were prepared at a total lipid concentration of 1 mM and AHR at 20 μM. The light scattering due to LUVs in suspension was corrected with the subtraction of an appropriate blank. 

### 2.5. Strains and Growth Conditions

The MRSA/Vancomycin-intermediate *S. aureus* (VISA) strain CIP 106760 (Institut Pasteur CIP) was used in this study, and was kindly given by Professor Aida Duarte from Universidade de Lisboa, Portugal. All experiments were performed in Mueller–Hinton (MH) broth agar at 37 °C, with aeration. 

### 2.6. Synergy Studies

The antibiotic synergy effects between AHR with methicillin, vancomycin and ampicillin were evaluated through the well-diffusion method. The synergy was determined by measuring the growth inhibition halos in MH agar plates. AHR (1 mg/mL) was combined with selected antibiotics (1 mg/mL), at a final concentration of 0.5 mg/mL. Plates were incubated at 37 °C for 24 h. DMSO, AHR and antibiotic growth inhibition halos were performed as controls. The study was conducted in at least three independent assays.

### 2.7. Bacterial Growth Curve

Strain CIP 106760 was grown in sterile non-treated 96-well microplates (Brand, Wertheim, Germany) in MH broth. Overnight cultures were diluted to an initial optical density (OD) OD_600nm_ of 0.05 and supplemented with AHR to final concentrations of minimum inhibitory concentration (MIC), MIC/2 and 2MIC. Cultures were grown at 37 °C with shaking (180 rpm) and monitored for 24 h in a microplate reader (Tecan Group Ltd., Männedorf, Switzerland). Three biological replicates were performed.

### 2.8. Effect of AHR on the Viability of CIP106760 Strain

To provide an estimate of viable counts, colony-forming units (cfu/mL) were determined. Mid-exponential cultures of CIP 106760 were supplemented with different concentrations of 7α-acetoxy-6β-hydroxyroyleanone (3.9 mg/L, 7.8 mg/L and 15.6 mg/L) and serial dilutions of the bacterial cultures were plated on tryptic soy agar (TSA, Difco) at discrete time points of the growth curve. The plates were incubated for 48 h at 37 °C, and the colonies were counted. All assays were performed in triplicate, and the results are representative of the average and standard error of the mean (SEM).

### 2.9. Cell Leakage Assay

A mid-exponential culture of the CIP 106760 strain was centrifuged and resuspended in 0.9 % sterile NaCl solution, to yield a final OD_620nm_ of 3. AHR was added at final concentrations of MIC/2, MIC and 2MIC, and the cell suspensions were incubated at 37 °C. The cell supernatants were monitored by measuring the OD_260nm_ for 16 h. All assays were performed in triplicate, and results are representative of the average and standard error of the mean (SEM).

### 2.10. Membrane Interaction and Leakage Assay

The AHR effect on lipid bilayers passive permeability was monitored by measuring the leakage of intraliposomal CF, through the concomitant increase of fluorescence intensity [8]. LUV suspensions were prepared in Hepes 10 mM, pH 7.4, containing CF at a concentration of 40 mM. Non-encapsulated CF was separated from the vesicles’ suspension through gel filtration in a Sephadex G-25 column. The fraction containing the LUVs with encapsulated CF was distributed in a 96-well standard opaque microplate to a final lipid concentration of 0.5 mM, in a final volume of 250 μL. Concentrations of MIC/2, MIC and 2MIC were tested. Liposome suspensions containing CF in the presence of 0.9 % of DMSO were used as control. 

The variation of fluorescence intensity was measured at excitation and emission wavelengths of 492 and 530 nm, respectively, with a cut off filter of 515 nm, using a microplate reader (Gemini EM Microplate Reader, Molecular Devices), at 25 °C. The fluorescence intensity measurements were performed initially only with the LUV suspension, then in the presence of different AHR concentrations, and finally, by adding Triton X-100 to a final concentration of 0.5 % (*v/v*), to obtain the value corresponding to the complete release of CF. The quantification of leakage was determined according to the following Equation 1 [8]:(1)$$Leakage\;\left(\%\right)=\frac{F_{p}-F_{0}}{F_{100}-F_{0}}\times 100$$
where *F_p_* corresponds to fluorescence intensity value over time, *F*_0_ is the initial fluorescence of the vesicle suspension, and *F*_100_ is the fluorescence intensity value after the addition of Triton X-100. The values were converted considering the control measurements.

The phospholipid concentration in the final LUV suspensions was confirmed by the quantification of inorganic phosphate following the colorimetric method of Rouser [9].

### 2.11. Proton Leakage Assay

The effect of AHR on ion exchange through lipid bilayers was followed by ratiometric fluorimetric measurements using a pH-sensitive probe, HPTS, following the procedures described in [7]. LUVs of POPC/PSM/cholesterol, in the proportions of 59.7:26.3:14 (26 mol% liquid ordered (*l_o_*) phase) and 34:32.7:33.3 (83 mol% *l_o_* phase), were prepared in citrate phosphate buffer (citric acid 100 mM, Na_2_HPO_4_ 200 mM, pH 5.0) containing 0.5 mM of HPTS. Those two- lipid composition are referred further ahead as low-cholesterol and high-cholesterol, respectively. The excess HPTS was separated from LUVs with encapsulated HPTS by gel filtration in a Sephadex G-25 column. 

The LUV suspension fraction was distributed into a 96-well standard opaque microplate to a final lipid concentration of 0.2 mM, in a final volume of 250 μL per well. AHR concentrations employed and the control were the same as in the CF leakage assay.

The pH measurements inside the liposomes were performed using the ratio of fluorescence at two excitation wavelengths, 405 and 450 nm, and a fixed emission wavelength of 510 nm (IF_450/405_), with a 495 nm cut off filter, in the microplate reader referred to above, at 25 °C. Finally, Triton X-100 was added to a final concentration of 1% (*v/v*) to acquire the intensity ratio corresponding to the total dissipation of the pH gradient. The dissipation percentage of the pH gradient (% ΔpH), considering as appropriate a linear relation with IF_450/405_ in the pH range covered, was determined according to Equation 1 [8].

### 2.12. Cell Surface Charge

The influence of AHR in bacterial surface charge was determined by cytochrome c binding, as described [1]. Bacterial cells were grown to a mid-exponential phase in the presence of AHR at MIC/2, MIC and 2MIC concentrations, recovered by centrifugation, and washed twice in phosphate buffered saline (PBS) buffer (pH 7.2). The cellular suspension was adjusted to an OD_600nm_ of 7.0 and was incubated with 0.5 mg/mL of cytochrome c for 10 min at room temperature; the supernatant was recovered by centrifugation and the cytochrome c content was determined by measuring the OD_530nm_. The experiment was performed in triplicate. The relative amount of bound molecule was determined by comparing the obtained values with the initial cytochrome c solution. All assays were performed in triplicate, and the results are representative of the average and standard error of the mean (SEM).

### 2.13. Lysis Assay

Heat-inactivated cells of MRSA/VISA strain were prepared as described in [11] and adjusted to an initial OD_600nm_ of 0.3. Lysis assays were performed in sterile nontreated 96-well microplates at 37 °C with shaking (80 rpm), and measured for 2 h. AHR at MIC/2, MIC and 2MIC concentrations were added to the wells at the beginning of the assay and the OD_600nm_ was measured at 30 min, 1 h and 2 h. All assays were performed in triplicate, and the results are representative of the average and standard error of the mean (SEM).

### 2.14. Analysis of Peptidoglycan Composition 

Isolation of the cell wall was performed as previously described [12,13]. Briefly, cells grown to mid-exponential phase were harvested by centrifugation, washed twice with cold double-distilled water, resuspended in 10% sodium dodecyl sulfate and boiled, to remove the cell wall-associated proteins. The cells were mechanically disrupted using 106 mm glass beads (Sigma) and the cell wall fragments were incubated with 0.5 mg/mL trypsin to degrade cell-bound proteins. Purified cell walls were washed and incubated with 49% hydrofluoric acid to remove teichoic acids. The purified peptidoglycan was washed with water several times, to remove all traces of hydrofluoric acid, and then lyophilized. Identical amounts of peptidoglycan were digested with mutanolysin (1 mg/mL; Sigma). The resulting muropeptides were reduced with sodium borohydride and separated by reverse phase-high performance liquid chromatography (RP-HPLC) using a Hypersil ODS (Runcorn Cheshire, UK) column (3 mm particle size, 25064.6 mm, 120 A ° pore size) and a linear gradient from 5 % to 30 % MeOH in 100 mM sodium phosphate buffer pH 2.5, at a flow rate of 0.5 mL/min, as described [12,14].

### 2.15. Scanning Electron Microscopy (SEM) Analysis

The imaging of bacteria treated with AHR was performed by SEM with a scanning electron microscope JEOL 5200LV, JEOL Ltd. (Tokyo, Japan). Bacteria were incubated overnight in Mueller-Hinton broth at 37 °C. Cells were harvested by centrifugation at 3000 rpm for 10 min and washed twice in PBS buffer pH 7.4. The remaining cells were resuspended in PBS buffer, the OD_600 nm_ was measured and the cell suspension was diluted to obtain a value of 2. AHR was added at different concentrations and to the cell suspension was incubated for 30 min at room temperature. The preparation of different samples for imaging was performed as described by He et al. [15] A positive control was considered with non-treated cells.

## 3. Results and Discussion

### 3.1. Extraction Optimization of AHR 

*Plectranthus* species are known for their predominance to produce diterpenes with royleanone scaffold [6]. To obtain the AHR in high amounts, the extraction optimization was carried out according to three different extraction methods. The content of AHR in all three extracts was evaluated through HPLC-DAD. The results are shown in Table 1. A higher extractive capacity was evident, using the maceration process with acetone as extraction solvent. Lower yields were obtained with ultrasonic extraction comparing to maceration extraction.

### 3.2. AHR does not Provide a Synergistic Effect with Cell Wall Antibiotics

Antimicrobial combination therapies are often used with the intention of providing broad-spectrum coverage and preventing the emergence of resistant strains [16]. Having this in mind, the synergic effects of AHR with other antibiotics active against Gram-positive bacteria (methicillin, ampicillin and vancomycin) were studied (Table 2). As expected, when testing methicillin and ampicillin without coupling AHR, no visible effect was seen on bacterial growth. Regarding the effect of the glycopeptide antibiotic vancomycin used to treat MRSA caused infections, a visible effect on MRSA bacterial growth was observed (inhibition zone of 17±1 mm), and thus, no significative synergetic effects were detected with the antibiotics in the study.

### 3.3. Effect of AHR on the Growth Rate of CIP106760 Strain

Royleanone was added at multiple values of the MIC for three different concentrations; twice the MIC (15.6 µg/mL), MIC (7.8 µg/mL) and half the MIC (3.9 µg/mL), to a culture of the MRSA strain CIP106760 (Figure 1). The growth profiles of the strain in the presence of AHR were compared to the profile in the absence of the compound. The growth rate of CIP106760 (0.0061 min^−1^) decreased approximately 10-fold (0.0005 min^−1^) at the MIC concentration of AHR, and growth was almost completely impaired at twice the MIC concentration. For the lower AHR concentration tested, half the MIC, the growth rate decrease was approximately three-fold (0.0021 min^−1^), and the OD value (0.8) attained in the stationary phase was approximately half the OD_620nm_ value of CIP106760 (1.6), suggesting a strong effect on cell viability.

### 3.4. Effect of AHR on the Viability of CIP106760 Strain

To estimate the viability and inquire about the bacterial properties of AHR at different concentrations (MIC/2, MIC and 2MIC), the antibacterial activity of AHR was evaluated by monitoring the number of viable cells (cfu/mL), in parallel with the culture turbidity, over time (Figure 2). The bacterial growth curve was followed for 300 min from the addition of the compound, to the beginning of the exponential phase at 90 min. For half the MIC concentration (MIC/2, 3.9 µg/mL), the reduction observed in the growth rate of the bacterial culture was not accompanied by a loss in viability. However, for higher compound concentrations, at the MIC and double MIC values (MIC, 7.8 µg/mL and 2MIC, 15.6 µg/mL, respectively), the steadiness of the culture turbidity was accompanied by a continuous decrease in cell viability, suggesting a loss of viability. 

### 3.5. Effect of AHR on the cell integrity of CIP106760 strain

To determine if the mode of action of AHR involves the disruption of cellular integrity, cell leakage assays were performed (Figure 3). CIP106760 cells were incubated with AHR at the MIC/2, MIC and 2MIC concentrations, and the supernatant of the cell suspension was analyzed at discrete time points for 16 h after the challenge. The monitoring of the optical density value of the cell supernatant at a wavelength of 260 nm (OD_260nm_) allows one to obtain a relative quantification of the nucleic acid content, a result of the leakage of the cytoplasmic content from compromised cells.

The endopeptidase lysostaphin was used as a positive control, as it specifically disrupts the pentaglycine bridge of the peptidoglycan of certain *Staphylococcus* species, leading to rapid cell lysis [16]. As a negative control, the cells were incubated with DMSO. All tested concentrations of AHR caused no significant cell disruption until 180 min of incubation, presenting OD_260nm_ values similar to the negative control and much lower than the positive control, for which cell lysis was observed. However, from 240 min of incubation onwards, a modest increase of the OD_260nm_ value of the supernatant of cells treated with any of the concentrations of AHR tested suggests that a moderate leakage of nucleic acids occurs, possibly resulting from a controlled disruption of the membrane and/or cell wall, or from cell death.

### 3.6. Assessment of AHR Cell Wall Lytic Activity

To determine if the AHR could directly mediate the hydrolysis of the cell wall of *S. aureus*, cells of CIP106760 strain were heat-inactivated by autoclaving, washed from growth medium and resuspended in PBS buffer. After the normalization of the optical density, the inactivated cell suspension was incubated with AHR at the MIC/2, MIC and 2MIC concentrations, and the optical density of the cell suspension was analyzed at discrete time points over 2 h after the challenge. The incubations of the heat-inactivated cell suspensions with lysostaphin and DMSO were used as positive and negative controls respectively, as before. All tested concentrations of AHR caused no significant cell degradation, as the optical density was unchanged along the incubation time. Lysostaphin treatment resulted in a rapid decrease of the optical density of the cell suspension, indicating efficient cell degradation, as expected (Figure 4).

### 3.7. Cell Morphological Alterations upon Challenge with AHR

The morphological changes of the surface of CIP106760 cells treated with AHR at the MIC and half the MIC concentrations were observed by SEM (Figure 5). The treatment of cells with the double MIC concentration of AHR did not allow one to perform SEM observations. During the cell attachment process to the support, intact cells were no longer visible. At MIC and MIC/2 concentrations, the images showed the formation of large clusters of cells (Figure 5C) and a lack of complete cell lysis, consistent with the previously described results on the lack of cell wall lytic activity (see Figure 4). However, some bacteria showed localized cell wall disruption (Figure 5D, arrows). Furthermore, most cells present an obvious deformation of their native structure, losing the coccoid shape and adopting a more elongated aspect. Moreover, the surface of the cells lost its native smooth aspect, which may be related with the production of an aggregation matrix, as in a biofilm, or the incorrect trimming of cell surface-associated polymers.

### 3.8. Effect of AHR on Cell Wall Synthesis and Peptidoglycan Composition

Although the cell integrity and heat-inactivated cells lytic assays demonstrated that the AHR bactericidal effect does not involve cell lysis, AHR treatment (once again tested at MIC/2, MIC and 2MIC concentrations) resulted in severe changes in the cell morphology and cell surface, which suggest that the AHR interferes with cell wall synthesis or with its major component, the peptidoglycan. To explore this hypothesis, whole cell wall was extracted and the peptidoglycan was purified from CIP106760 and CIP106760 treated with AHR MIC/2, MIC and 2MIC concentrations. Significant differences were observed in the amount of total cell wall retrieved from the different cell cultures. The cell wall dry weight per gram of cell wet weight was 27.26 (±5.4) mg for CIP106760 and decreased continuously as the AHR concentration increased; 22.08 (±4.8) mg, 15.36 (±3.9) mg and 9.12 (±2.7) mg for CIP106760 grown in the presence of 3.9 µg/mL, 7.8 µg/mL and 15.6 µg/mL of AHR, respectively. These results demonstrate that the cell wall thickness of the cell must be drastically reduced in the presence of AHR. Regarding the peptidoglycan recovery yield, it represented 40–50% of the total cell wall amount, for all the conditions tested. However, for AHR concentrations of MIC and 2MIC, we detected a significant increase in the relative amounts of some muropeptide structures; corresponding peaks were identified in Figure 6 as peaks “a”, “b”, “c” and “d”. By comparing the HPLC profiles with previously reported studies, in which the muropeptide structures were identified by mass spectrometry [12], the muropeptides of peaks “a”, “b”, “c” and “d” correspond to the monomeric pentapeptide without the pentaglycine bridge, the monomeric pentapeptide with only a tetraglycine bridge, the dimeric pentapeptide with one pentaglycine bridge and one tetraglycine bridge, respectively.

These results suggest a relative increase in muropeptide structures that usually exist in very small amounts in the native *S. aureus* peptidoglycan. All these structures have less pentaglycine bridges or incomplete bridges, suggesting the inhibition of one, or more than one, of the steps of peptidoglycan biosynthesis that are responsible for the bridge formation. Such biosynthetic steps are mediated by FemA, FemB and FemX aminoacyltransferases that catalyze the sequential addition of the glycine residues to the lipid linked muropeptide, in a membrane-associated reaction.

### 3.9. Assessment of AHR Effect on the Cell Surface Net Charge

Cytochrome c is a cationic protein that may be used to estimate the relative surface charge of the cell envelope of *S. aureus* strains [17]. Due to its cationic nature, this protein associates to the cell surface that presents a negative net charge. In this way, as the association level of cytochrome c to bacteria surface is reduced, the higher will be the relative positive surface charge of the cell [18]. 

The results obtained (Figure 7) for the cells treated with 2MIC concentration of AHR showed a significantly decreased association of this externally added cationic protein to the bacteria surface, suggesting a drastic increase of the usual negative surface charge of the bacterial strain. Accordingly, AHR at such a concentration is able to affect the cell envelope. However, the inability to significantly alter the surface charge at the MIC suggests that this is a crucial step in the AHR mode of action.

From the images obtained with the scanning electron microscope, some bacteria showed localized cell wall disruption (Figure 5D, arrows) when exposed to the MIC concentration of AHR. It is possible that at the 2MIC concentration of AHR, this effect is enhanced and results in membrane depolarization.

### 3.10. Assessment of AHR Interaction and Effect on Membrane Lipid Bilayers’ Permeability

To assess the mechanism of action of AHR at the bacterial membrane level, the interaction of this antimicrobial diterpene with artificial lipid bilayers was carried out. DOPC at room temperature spontaneously forms a disordered phospholipid bilayer, representative of the fluid bilayers that are found in living organisms, with the additional advantages that it is available with very high purity and presents a very low turbidity compared to other phospholipids. This fact allows a complete correction of the absorption spectra. Since AHR has a good chromophore, the interaction between the compound and the lipid membrane can be easily tested using the absorption bands of the antibacterial compound. Thus, the interaction of AHR with DOPC bilayers in suspension was first studied by UltraViolet (UV)-visible absorption spectroscopy.

The electronic absorption spectrum of AHR in aqueous buffer (Figure 8A) showed one band in the near UV region with maximum at a wavelength of 275 nm, and a broad band in the visible region centered at a wavelength of 520 nm, which is responsible for the yellow color of the compound aqueous solution buffered at pH 7.4. In the presence of LUV, changes in the absorption bands were clearly observed. On one hand, the lipid bilayer induced hyperchromism of the band at 275 nm, possibly accompanied by a small hypsochromic shift. On the other hand, the band centered at ~520 nm underwent a hypochromic effect. The observed differences clearly indicate that AHR is surrounded by a less polar environment in the presence of 1 mM LUV, i.e., a significant fraction of the compound partitions into the membrane, where it becomes less accessible to highly polar water molecules [19].

After confirming that AHR interacts with DOPC bilayers at typical lipid concentrations and at relevant concentrations of the compound, its effect on membrane stability was tested. The effect of AHR on membrane leakiness was studied in lipid bilayers composed by DOPC and monitored by measuring the increase in fluorescence intensity as a consequence of the release of encapsulated CF. The release of this probe may occur as a consequence of the strong perturbation of membrane order, and other events, such as (hemi)fusion, pore formation and detergent-like action. The representative curves of CF leakage kinetics from DOPC LUV suspension in the presence of different concentrations of AHR are shown in Figure 8B. The presence of AHR at MIC and 2MIC led to ~10% of CF maximum leakage. At the compound: at a lipid molar ratio of 1:25 (MIC), AHR had a small effect in the passive permeability of lipid bilayers, when compared, e.g., with a fusion peptide SARS_FP_ which induced ~60% CF leakage for the same 1:25 molar ratio [8]. Furthermore, no differences between 2MIC and MIC concentrations of AHR were detected. AHR at the MIC/2 presented at the most ~3% of CF leakage, which is quite low compared to the effect of SARS_FP_ (~50% CF leakage) [8] and antimicrobial peptides (~90–100% CF leakage) [20] at the same compound: lipid molar ratio of 1:50. In summary, AHR does not affect the integrity of fluid membranes.

Since CF is a small organic molecule, it would be possible that AHR induced the formation of very small pores or defects that would not be stable or large enough to allow the release of the CF molecule, but would lead to the dissipation of ionic gradients. Therefore, the effect of AHR on membrane passive permeability to protons was evaluated through the variation of the fluorescence intensity ratio of encapsulated HPTS, a pH-sensitive probe, at two excitation wavelengths (Figure 9). The liposomes were subjected to a proton gradient, i.e., the pH of the internal solution was 5.0 and the external pH was 7.4 [7]. AHR did not greatly influence the passive permeability to protons in both model systems used. In fact, only ~5% and ~1.5% of the pH gradient dissipation for bilayers with low and high concentration of cholesterol, respectively, were obtained.

Moreover, no significant differences between the different AHR concentrations used were observed. Moreover, the presence of DMSO led to a small % ΔpH, allowing one to conclude that DMSO does not affect passive permeability, and the effect of AHR is almost negligible in comparison to the control (Table 3). These results, obtained with the low cholesterol liposomes which resemble, in biophysical properties, mammalian intracellular membranes or bacterial membranes, are concomitant with those obtained in CF leakage assays. Thus, AHR at MIC did not show a noticeable effect on passive permeability and, as stated above, AHR did not disturb the membrane structure. Furthermore, regarding the results presented above, the AHR effect is even smaller for the lipid mixtures containing a high concentration of cholesterol, which mimics the plasma membrane of mammalian cells. This result is consistent with the fact that AHR displays a low cytotoxicity towards human cells (GI_50_ 12.80 µg·mL^−1^) [2].

An integration of all the results obtained leads us to suggest that the mode of action of AHR shows some resemblances with that of daptomycin, that is not yet fully elucidated. As for daptomycin, AHR interacts with the membrane, does not result in cell lysis, and alters the homeostasis of the cell wall [21]. Additional assays will be performed to compare AHR and the daptomycin effect on *S. aureus*, to enable one to further elaborate on the similarities between the mode of action of these two compounds.

## 4. Conclusions

AHR is not only an effective antibacterial agent, as it also exerts promising activity against the multidrug resistant MRSA/VISA strain CIP 106760. Having in mind that the purpose of this study was to unveil its mechanism of action, it seems that this abietane diterpene does not affect the bacterial cell membrane, but rather its cell wall. However, its ability to interact with a fluid phospholipid membrane without significantly perturbing it suggests that AHR is able to permeate the bacteria cell membrane and exert intracellular actions, which may result in some of the alterations observed at the cell wall level.

The results showed that AHR is able to disrupt the cell wall without causing the lysis of the bacteria.

The herein presented results suggest that additional studies should be made, in order to further assess the AHR antibacterial effect, toxicity and possible clinical use. Additionally, the studies will be widened to a representative collection of clinical strains that cover the main *S. aureus* genetic lineages, illustrative of the genetic background heterogeneity presented by this species [22].

## Figures and Tables

**Figure 1 biomolecules-10-00983-f001:**
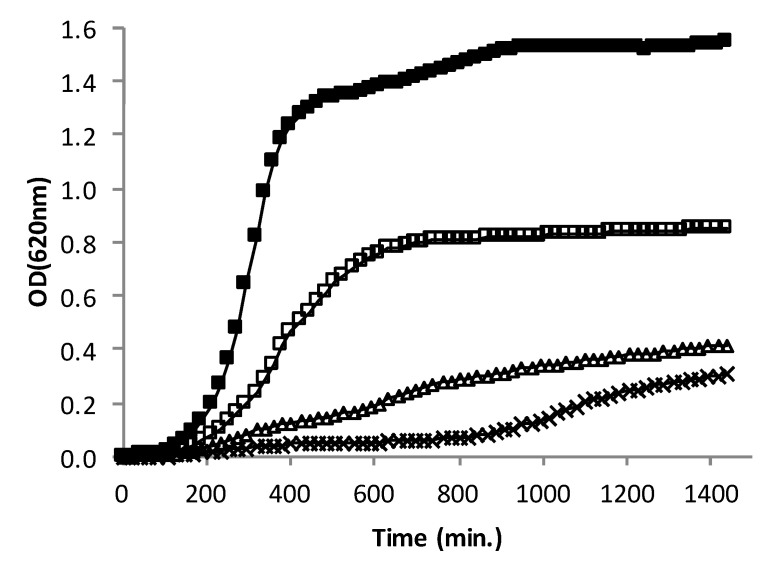
Representative growth curves of MRSA CIP106760 (■) and CIP106760, challenged with different concentrations of 7α-acetoxy-6β-hydroxyroyleanone; (□) half the MIC (3.9 mg/L); (Δ) MIC (7.8 mg/L); (✖) double the MIC (15.6 mg/L). Growth was monitored by measuring the OD_620nm_ for 24h in a microplate reader.

**Figure 2 biomolecules-10-00983-f002:**
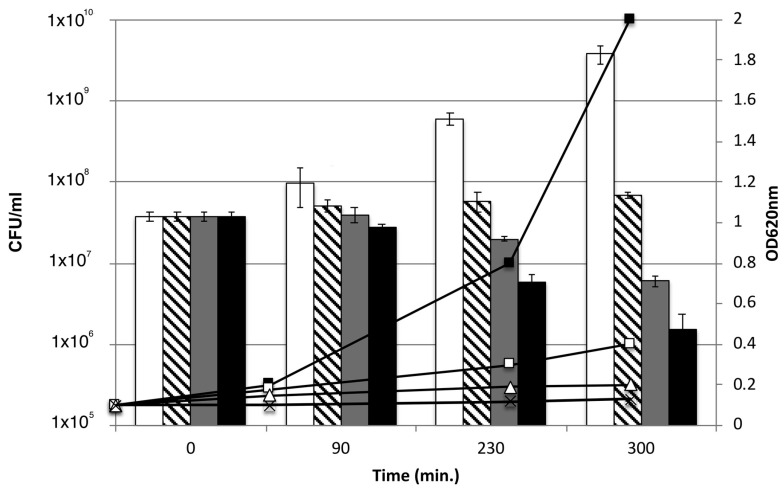
Comparison of the effect of AHR on the culture turbidity and on the cell viability. The optical density (OD_620nm_) and Colony-forming unit (cfu)/mL were measured at discrete time points and represented as a line chart and a column chart, respectively. The growth profiles were determined for MRSA CIP106760 (■; white bar) and CIP106760 was challenged with different concentrations of 7α-acetoxy-6β-hydroxyroyleanone; (□; striped bar) MIC/2 (3.9 mg/L); (Δ; grey bar) MIC (7.8 mg/L) and (✖; black bar) 2MIC (15.6 mg/L). Experiments were performed in triplicate. Shown are means with the standard error.

**Figure 3 biomolecules-10-00983-f003:**
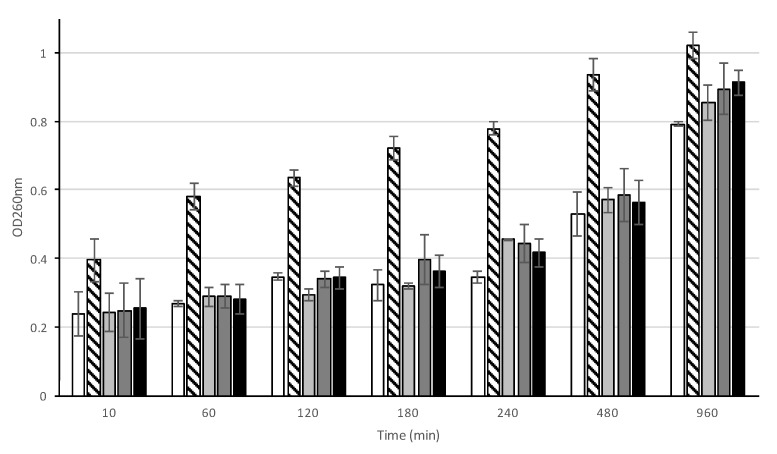
Cell leakage assay of MRSA CIP 106760 strain challenged with 7α-acetoxy-6β-hydroxyroyleanone. The optical density (OD_260nm_) of the cell supernatant was measured at discrete time points for MRSA CIP106760 in the presence of DMSO (white bar), CIP106760 challenged with 100 mg/L of lysostaphin (striped bar) and CIP106760 challenged with different concentrations of 7α-acetoxy-6β-hydroxyroyleanone; MIC/2 (3.9 mg/L; light grey bar); MIC (7.8 mg/L; dark grey bar) and 2 MIC (15.6 mg/L; black bar). Experiments were performed in triplicate. Shown are means with the standard error.

**Figure 4 biomolecules-10-00983-f004:**
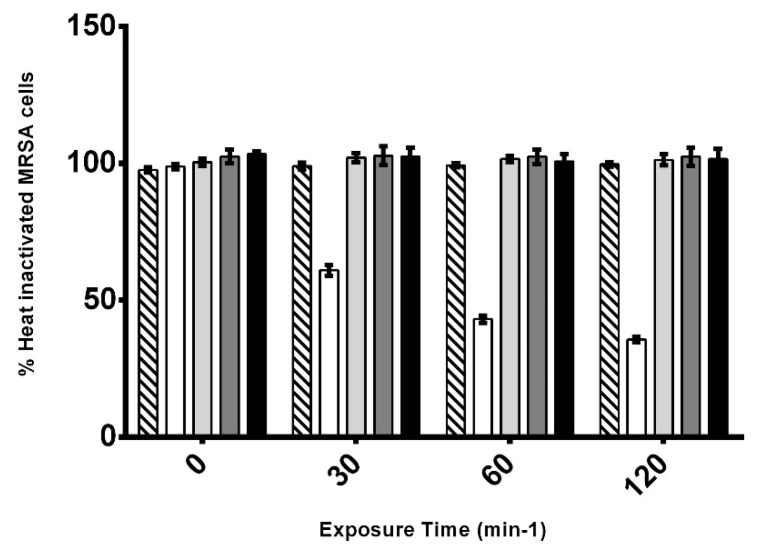
Lysis assay of heat-inactivated cells of MRSA CIP 106760 strain, with different concentrations of 7α-acetoxy-6β-hydroxyroyleanone. The optical density (OD_600nm_) of the cell suspensions was measured at discrete time points for the effect of DMSO (black bar), 100 mg/L of lysostaphin (white bar) and different concentrations of ARH; MIC/2 (3.9 mg/L; medium grey bar); MIC (7.8 mg/L; light grey bar) and 2 MIC (15.6 mg/L; dark grey bar). Experiments were performed in triplicate. Shown are means with the standard error.

**Figure 5 biomolecules-10-00983-f005:**
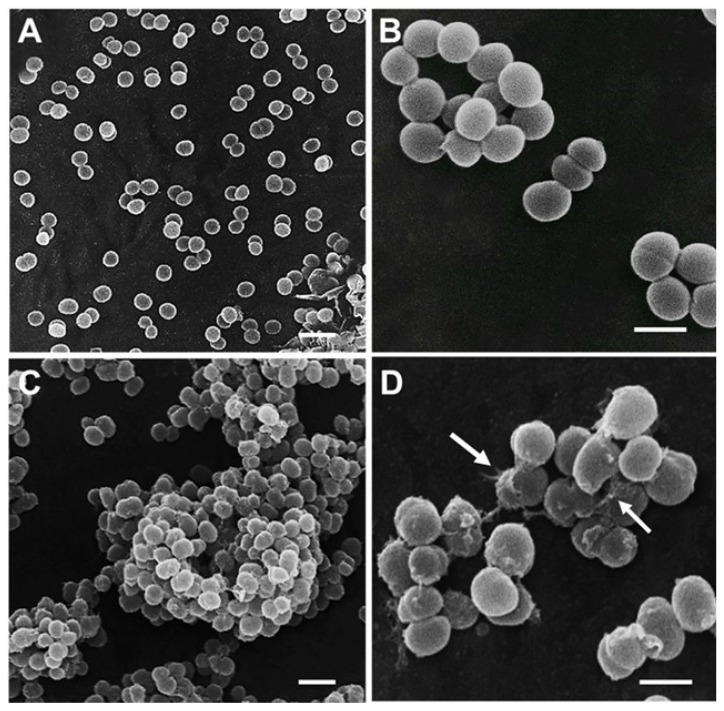
SEM micrographs of the MRSA/VISA CIP 106760 strain. **A**, **B**—Control bacteria. **C**—Bacteria treated with AHR at MIC/2, **D**—Bacteria treated with AHR at MIC showing localized cell wall disruption (arrows). Scale Bars: 4 μm (A, C); 1 μm (B, D).

**Figure 6 biomolecules-10-00983-f006:**
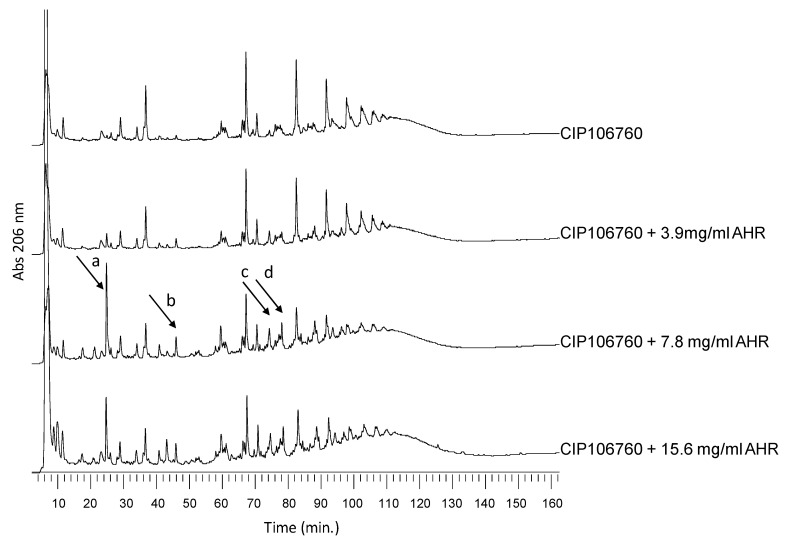
RP-HPLC peptidoglycan profiles. The purified peptidoglycan was digested with mutanolysin, reduced and analyzed by RP-HPLC. Muropeptide profiles of CIP106760 strain challenged with MIC/2, MIC and 2MIC concentrations of AHR. Muropeptide structures corresponding to peaks a, b, c and d (highlighted by arrows) are over-represented in the peptidoglycan of CIP106760 challenged with MIC and double MIC concentrations of AHR.

**Figure 7 biomolecules-10-00983-f007:**
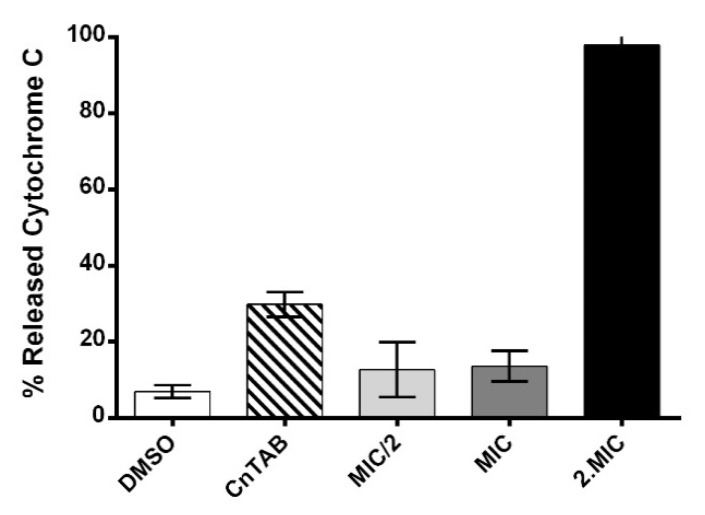
Percentage of cytochrome c that was not associated to the surface of S. aureus cells in the presence of AHR at different concentrations. CnTAB—positive control. Experiments were performed in triplicate. Shown are means with the standard error.

**Figure 8 biomolecules-10-00983-f008:**
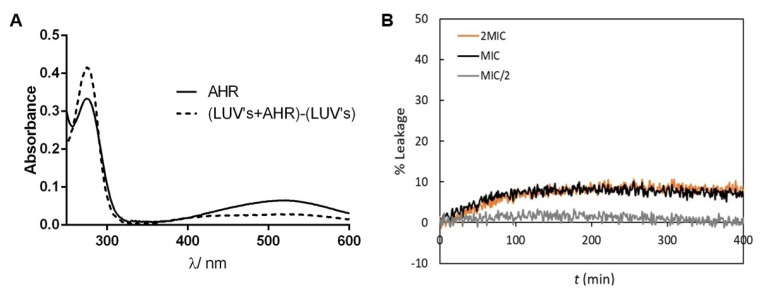
AHR at relevant antimicrobial concentrations interacts with phospholipid bilayers without compromising their integrity. (**A**) Electronic absorption spectrum of AHR at 20 µM in buffer Tris-HCl, pH 7.4 in the absence and presence of 1mM 1,2-dioleoyl-*sn*-glycero-3-phosphocholine (DOPC) large unilamellar vesicles (LUV) suspension. (**B**) Representative curves (median behavior from 6 independent experiments) of the effect of AHR on CF leakage from DOPC LUVs in suspensions, with a total lipid concentration of 0.5 mM, in the presence of different AHR concentrations: 15.6 mg/L (2 MIC, 40 µM), 7.8 mg/L (MIC, 20 µM) and 3.9 mg/L (MIC/2, 10 µM). These experiments were performed at 25 °C.

**Figure 9 biomolecules-10-00983-f009:**
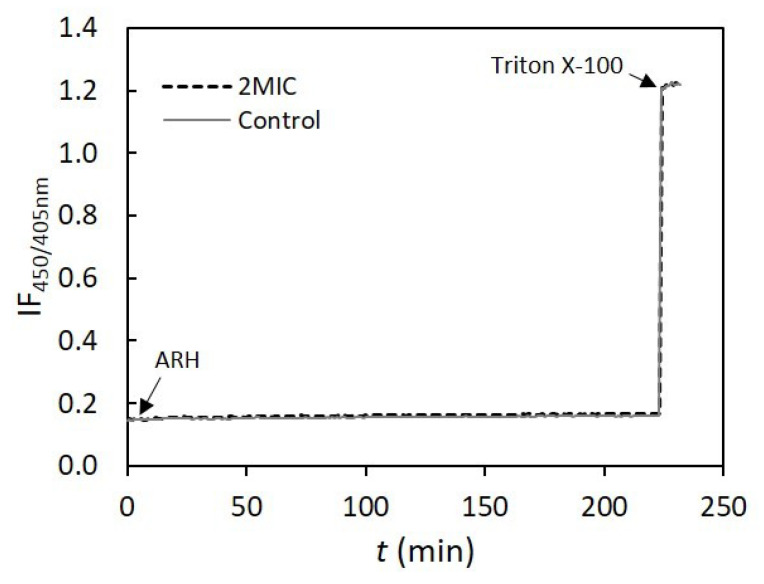
Effect of AHR at 15.6 mg/L (2 MIC, 40 µM) on lipid bilayer passive permeability to protons. Representative curves of the time-dependence of fluorescence intensity ratio of 8-hydroxypyrene-1,3,6-trisulfonic acid (HPTS) at the excitation wavelengths of 405 and 450 nm (IF_450/405_), at 25 °C. The experiments were conducted with high-cholesterol LUVs at total lipid concentration of 0.2 mM. Triton X-100 (1%) was added at 220 min, to obtain the value of IF_450/405_ corresponding to total pH gradient dissipation.

**Table 1 biomolecules-10-00983-t001:** Quantification of 7α-Acetoxy-6β-hydroxyroyleanone (AHR) in different *P. grandidentatus* extracts.

Extraction Method	AHR Index (µg mg^−1^ Extract)
Acetone maceration	73.3
Acetone ultrasonic-assisted	52.0
Supercritical fluid	52.3

**Table 2 biomolecules-10-00983-t002:** Synergy effect of AHR and three different antibiotics against the methicillin-resistant *Staphylococcus aureus* (MRSA)/Vancomycin-intermediate *S. aureus* (VISA) CIP 106760 strain.

Compound.	Zone Inhibition (mm).			
	DMSO	MET	AMP	VANC
**AHR**	21 ± 2	20 ± 2	20 ± 2	19 ± 3
**No AHR**	5	5	5	17 ± 1

MET—methicillin; AMP—ampicillin; VAN—vancomycin.

**Table 3 biomolecules-10-00983-t003:** Effect of different concentrations of AHR on lipid bilayer passive permeability to protons monitored through HPTS ratiometric fluorimetry, at 25 °C. The experiments were conducted in lipid mixtures containing low and high concentrations of cholesterol. The percentage of pH-gradient dissipation (% ΔpH) presented is related to the outward movement of protons trough the lipid bilayer before the total dissipation of pH gradient. The values are the mean ± S.D. of three independent experiments.

AHR Concentration	% ΔpH	
	Low Cholesterol LUVs	High Cholesterol LUVs
**15.6 mg/L (2MIC)**	5.1 ± 0.1	1.4 ± 0.2
**7.8 mg/L (MIC)**	4.8 ± 0.3	1.5 ± 0.3
**3.9 mg/L (MIC/2)**	4.6 ± 0.7	1.6 ± 0.2
**Control**	3.6 ± 0.2	1.3 ± 0.1

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
