# Peer review of "Unveiling the Mechanism of Action of 7α-acetoxy-6β-hydroxyroyleanone on an MRSA/VISA Strain: Membrane and Cell Wall Interactions"

_biomolecules, 2020, doi:10.3390/biom10070983_

Round 1

Reviewer 1 Report

The athors describe the antimicrobial activity of a diterpene i.e. 7α-Acetoxy-6β-hydroxyroyleanone (AHR),

The authors describe the antimicrobial activity of a diterpene i.e. 7α-Acetoxy-6β-hydroxyroyleanone (AHR), against methicillin-resistant Staphylococcus aureus (MRSA) strains.

The authors state that the compound disrupts the bacterial cell wall without causing the lysis of the bacteria.

Despite the paper contributes to the field of natural products and their properties, substantial revisions need to be done.

Please check the english and typing mistakes. Can the authors prove that the membrane perturbation is the cause of bacterial killing? The kinetics of these two events should overlap. The Sytox Green assay to monitor the bacterial membrane perturbation in real time would be helpful to this aim (see for example PMID: 16566601)

How can the authors exclude that DNA release is not a consequence of bacterial death?

Line 488 remove REF.

In the introduction the authors may include recent publications on the antimicrobial activity of diterpenes (see for example PMID: 32435382)

A paragraph on the statistical analysis of data should be included.

Author Response

Manuscript: Unveiling the mechanism of action of 7α-acetoxy-6β-hydroxyroyleanone on an MRSA/VISA strain: membrane and cell wall interactions

Dear Editor,

We thank and appreciate the reviewers’ comments, which have helped us to improve the manuscript. We have carefully considered the suggestions, addressing and incorporating them in the manuscript as detailed below. The modifications in the corrected paper in track changes and also have a Green highlight for reviewer.

Reviewer #1: Comments to the Authors

The authors describe the antimicrobial activity of a diterpene i.e. 7α-Acetoxy-6β-hydroxyroyleanone (AHR). The authors describe the antimicrobial activity of a diterpene i.e. 7α-Acetoxy-6β-hydroxyroyleanone (AHR), against methicillin-resistant Staphylococcus aureus (MRSA) strains. The authors state that the compound disrupts the bacterial cell wall without causing the lysis of the bacteria. Despite the paper contributes to the field of natural products and their properties, substantial revisions need to be done.

Comment 1: Please check the english and typing mistakes. Can the authors prove that the membrane perturbation is the cause of bacterial killing? The kinetics of these two events should overlap. The Sytox Green assay to monitor the bacterial membrane perturbation in real time would be helpful to this aim (see for example PMID: 16566601). How can the authors exclude that DNA release is not a consequence of bacterial death?

Authors: Thank you for the suggestions of the reviewer. The English and typing mistakes were corrected in the all manuscript. Considering the Sytox Green assay in the PMID: 16566601 (Effect of Natural L- to D-Amino Acid Conversion on the Organization, Membrane Binding, and Biological Function of the Antimicrobial Peptides Bombinins H) this seams an interesting assay used in a different case compared to our work that we could use in our following works.

The purpose of the cell leakage analysis was to provide preliminary results on cell leakage to direct the next assays to be performed in order to elucidate the mode of action of the compound and if it involves the disruption of cell integrity. Preliminary data from this assay suggested that no cell leakage occurred during the first 180 min, but some partial cell leakage could be occurring for longer times, suggesting a slight cell leakage effect. This result prompted us to perform more sensitive assays of AHR membrane interaction and its effect on membrane lipid bilayers permeability (see topic: “Assessment of AHR interaction and effect on membrane lipid bilayers permeability”). These results prompted us to suggest that the DNA release was due to a controlled disruption of the membrane and/or cell wall, but we cannot exclude this release is not the consequence of cell death. This comment was added in line 310.

Comment 2: Line 488 remove REF.

Authors: We thank the correction of the reviewer, we remove the REF in the text which was a mistake.

Comment 3: In the introduction the authors may include recent publications on the antimicrobial activity of diterpenes (see for example PMID: 32435382)

Authors: We thank the reviewer comment. Indeed we include the information of Diterpenes and their chemical structures are important to antimicrobial activity and add this paper in the references.

Comment 4: A paragraph on the statistical analysis of data should be included.

Authors: We thank the suggestion which took us to include a paragraph on the statistical analysis of the data as appropriate in the materials and methods section (2.8, 2.9, 2.12, 2.13) “All assays were performed in triplicate, and results are representative of the average and standard error of the mean (SEM).”

Reviewer 2 Report

The manuscript as it stands now reads well and is a significant contribution to a important field

Author Response

Thank you for the reviewer comment.

Reviewer 3 Report

This study is designed to show the antibacterial effect of 7α-acetoxy-6β-hydroxyroyleanone (AHR) against MRSA in vitro. Pereira et al. showed that AHR has bactericidal activity against MRSA, and further study was investigated to clarify the mechanism of action of AHR on MRSA. Interestingly, the results showed that AHR does not have a significant effect in membrane passive permeability, despite AHR interacting with lipid bilayers of the bacterial cell membrane. From these results, the antibacterial effect of AHR seems to be clearly different from that of Lysostaphin. However, the present study does not provide evidence that revealed the newly bactericidal mechanisms of AHR. Therefore, there is no provision of sufficient evidence to achieve the study aim and this appears to be a preliminary study. If additional experiments are examined and the newly antibacterial mechanisms of AHR are revealed, these results will have a higher impact on this research field. Specific comments are described as follows.

1) Are the value of MIC and the activity of Cytochrome C releasing of Lysostaphin against MRSA-CIP106760 stronger than AHR? It is better to discuss whether the antibacterial activity is characterized by the difference in the mechanism.

2) It is a wonder the difference between the antibacterial activity between Lysostaphin and AHR. In Figure 5, it needs to add the data of Lysostaphin-treated MRSA to compare with that of AHR-treated MRSA.

3) Line 70-71, correct the sentence "is herein studied. is an essential field for further study."

Author Response

Manuscript: Unveiling the mechanism of action of 7α-acetoxy-6β-hydroxyroyleanone on an MRSA/VISA strain: membrane and cell wall interactions

Dear Editor,

We thank and appreciate the reviewers’ comments, which have helped us to improve the manuscript. We have carefully considered the suggestions, addressing and incorporating them in the manuscript as detailed below. The modifications in the corrected paper in track changes and also have a Green highlight for reviewer.

Reviewer #3: Comments to the Authors

This study is designed to show the antibacterial effect of 7α-acetoxy-6β-hydroxyroyleanone (AHR) against MRSA in vitro. Pereira et al. showed that AHR has bactericidal activity against MRSA, and further study was investigated to clarify the mechanism of action of AHR on MRSA. Interestingly, the results showed that AHR does not have a significant effect in membrane passive permeability, despite AHR interacting with lipid bilayers of the bacterial cell membrane. From these results, the antibacterial effect of AHR seems to be clearly different from that of Lysostaphin. However, the present study does not provide evidence that revealed the newly bactericidal mechanisms of AHR. Therefore, there is no provision of sufficient evidence to achieve the study aim and this appears to be a preliminary study. If additional experiments are examined and the newly antibacterial mechanisms of AHR are revealed, these results will have a higher impact on this research field. Specific comments are described as follows.

Comment 1: Are the value of MIC and the activity of Cytochrome C releasing of Lysostaphin against MRSA-CIP106760 stronger than AHR? It is better to discuss whether the antibacterial activity is characterized by the difference in the mechanism.

Authors: We acknowledge the reviewer for his commentary. The anti-staphylococcal bactericidal activity of the enzyme Lysostaphin is due to its glycyl- glycine endopeptidase activity. Lysostaphin results in cell lysis through cleavage of the pentaglycine bridge of the peptidoglycan structure, which is specific to Staphylococcus species. The mechanism of action of lysostaphin and AHR are radically different, as lysostaphin is known to result in full cell lysis through an enzymatic digestion activity. Although in both cases, the pentaglycine bridge is involved in the mode of action, while lysostaphin actively cleaves the pentaglycine bridge, AHR action results in an altered cross-linking bridge but not in its degradation. Lysostaphin was chosen as a comparison control to address the capacity of AHR to mediate cell lysis, a feature that was not observed. Due to its fast lytic activity of lysostaphin, the surface charge assay (cytochrome C) is not applicable since the lysed cell debris are not separable from the soluble cytochrome C by centrifugation.

Comment 2: It is a wonder the difference between the antibacterial activity between Lysostaphin and AHR. In Figure 5, it needs to add the data of Lysostaphin-treated MRSA to compare with that of AHR-treated MRSA.

Authors: We thank the reviewer comment. the imaging analysis of cells of Staphylococcus aureus treated with lysostaphin is well documented in the literature “Ceotto-Vigoder H., S.L.S. Marques, I.N.S. Santos, M.D.B. Alves, E.S. Barrias, A. Potter, D.S. Alviano and M.C.F. Bastos. 2016. Nisin and lysostaphin activity against preformed biofilm of Staphylococcus aureus involved in bovine mastitis. J. Applied Microbiol. 121, 101-114”, “Watters C.M., T. Burton, D.K. Kirui, N.J. Millenbaugh. Enzymatic degradation of in vitro Staphylococcus aureus biofilms supplemented with human plasma. 2016. Infection and Drug Resistance. 9. 71–78” and “Francius G., O. Domènech, M. Mingeot-Leclercq, Y.F. Dufrêne. Direct Observation of Staphylococcus aureus Cell Wall Digestion by Lysostaphin. 2008. J. bacterial. 190(24):7904-9”. From these works, it is clear that at low concentrations, lysostaphin treated cells become osmotically fragile and at higher concentrations, the treated cells show completely altered morphology, compatible with full cell wall disruption.

Comment 3: Line 70-71, correct the sentence "is herein studied. is an essential field for further study."

Authors: We thank the reviewer comment that helps us to correct the manuscript.

Round 2

Reviewer 1 Report

The authors properly answered to my raised questiosn

Reviewer 3 Report

OK, no further comment and question.

This manuscript is a resubmission of an earlier submission. The following is a list of the peer review reports and author responses from that submission.

Round 1

Reviewer 1 Report

Fix the sentence structure line 44-45. The phrase 'is also an important field of action with great matter.' could be better expressed 'is an essential field for further study' as a suggestion. line 72 and again line 75, 1 l should be 1 L (Litre = L) line 199 states extract outcomes can be seen in table 1. This is not the case, table 1 contains MIC, MIC/2 and 2MIC concentration but not the crude extract weights or % as indicated. Change wording or change table to reflect statement suggest formatting change in Figure 3 and 4 description could make it clearer. '(light grey bar) MIC/2 (3.9 mg/L)' would be clearer as MIC/2 (3.9 mg/L; light grey bar) as an example Figure 5 does not show SEM results as indicated line 292 and 294, it appears the incorrect image has been displayed. This should be Figure 6 and subsequently Figure 6 is also out of line. Therefore Figures from 5 onward need to be revised, Figure 5 is missing Line 230 states results suggest bacteriostatic action, line 307 states bactericidal action, conflicting mode of action outcome. earlier comment should be revised as results indicate bactericidal action Section 3.8 needs a clearer discussion around the reported results and the Figure, specifically what this actually means. Reviewer read this section four or five times and still felt unconvinced of what the significance was of the 2MIC result compared with MIC. What this means in regard to measurable outcome The insertion of MOA resembling that of Daptomycin in the conclusion should be reconsidered. The statement is not entirely wrong but broad in its statement, therefore not entirely correct either regarding actual MOA and out of line with fact and the reported results see https://doi.org/10.3390/antibiotics9010017

Reviewer 2 Report

The manuscript describes mode of action studies of a diterpenoid AHR towards a multidrug-resistant S. aureus strain. A spectrum of different assays have been performed which gives good merit for this attempt to understand the biology underlying the observed antibacterial effect. However, I have a few major concerns that must be addressed before considering the paper for publication:

  1. Two methods for isolating the target compound AHR from the plant material are described, but the manuscript lacks any information on the purity of the compound or means for its identification. Any impurities may remarkable affect the bioassay results and they authors must provide data on compound purity and possible residual solvents.
  2. A main conclusion in the study is that AHR does not cause bacterial cell lysis. The leakage assay used to study bacterial cell membrane integrity is based on OD measurement of the extracellular medium and is very insensitive. More sensitive assays based on eg. beta-galactosidase or other enzyme leakage are widely applied and should be used in order to make statements in this respect.
  3. The conclusions are not fully in line with the presented data. For example, the authors state in the conclusions section and in the end of abstract that the antibacterial activity of AHR resembles that of daptomycin, yet daptomycin was not used in this study. The possible similarities may certainly be discussed within the text but this is not a conclusion based on the presented data.

In addition, I wish to draw the authors' attention to the following aspects:

4. The abstract should describe the results of the study in more detail and shorten the general introduction. In particular, do not present numberic data not originating from the current study (eg. cytotoxicity / selectivity data). Remove also the comparison to daptomycin from the abstract.

5. Include the supplier of the bacterial strains in Materials and Methods.

Reviewer 3 Report

The manuscript contains some interesting concepts and experimental designs. However, the manuscript is not without major concerns.

A single strain of MRSA is used for a multitude of different types of studies. While some of the research design is note-worthy and valuable, much of it is subject to grave concerns. Remember, however, that there is grave concern throughout every experiment reported which is that a single strain of MRSA was used. There is absolutely no way that conclusions can be drawn from an n=1 which makes the results totally unable to be generalized to larger population and to advancing the field of science and ultimately medical care for patients with MRSA infections. Generally speaking, grandiose conclusions are being made by the authors from results of a single organism.

Experimental concerns exist also.

Several experiments were done where bacterial growth was density standardized and incubated and then tested for effect. A major concern is that the density standardization is different for each experiment and lengths of time of incubation with ½ MIC, MIC and 2MIC is different. There seems to be no reason for how each experiment was performed and how one experiment linked with another one. It was as if the authors had no hypothesis or model system that they were testing. For example, the bacterial growth curve experiment was done at OD=0.05 and 24-hour incubation; cell leakage was assessed for OD=3 and 4-hour incubation; cell surface charge analysis was performed at OD=7 and 10-minute incubation; lysis was assessed at OD=0.3 and 2-hour incubation; and SEM analysis was done at OD=2 with 30-minute incubation. Interpretation of results is impossible with bacterial numbers and kinetics of studies so vastly different.

Many results in the Results Section text and displays are either missing text and/or figures or are very difficult to understand and interpret. For example, the text for Table 1 are not really what is displayed in Table 1. Actually, Table 1, as written up in the text, is in the Supplement Section. An experiment was not written up in the Methods Section (viability results: CFU/ml) but results were displayed in Figure 2. Results displayed are not always interpreted correctly. For example, the authors claim that the 300-minute timepoint in Figure 2 “shows the beginning of the exponential phase”: actually the beginning of that phase starts at around 90 minutes. In several Figure Legends, the authors claim n=3 which is totally incorrect. They tested a single strain of MRSA which is an n=1. Their claim of n=3 may be their thinking that “replicates” and “n” are the same thing.

The authors get off track in their numbering of Figures when they claim in text in the Results Section that Figure 5 shows SEM results. Indeed, Figure 5 results are of peptidoglycan results and SEM results are displayed in the supplement. From this point on, the Figure are mis-numbered.

Finally, their experiment using cytochrome c as an indicator of surface charge may be interesting but there is concern that what is being measured may be other functional consequences of the reaction. Sufficient controls were not included to alleviate these concerns.